# Introducing Less-Invasive Surfactant Administration into a Level IV NICU: A Quality Improvement Initiative

**DOI:** 10.3390/children8070580

**Published:** 2021-07-07

**Authors:** Steven M. Conlon, Allison Osborne, Julie Bodie, Jaime Marasch, Rita M. Ryan, Tara Glenn

**Affiliations:** Rainbow Babies & Children’s Hospital, Case Western Reserve University, Cleveland, OH 44106, USA; Steven.Conlon@uhhospitals.org (S.M.C.); Allison.Osborne@uhhospitals.org (A.O.); Julie.Bodie@uhhospitals.org (J.B.); Jaime.Marasch@uhhospitals.org (J.M.); Tara.Glenn@uhhospitals.org (T.G.)

**Keywords:** bronchopulmonary dysplasia, surfactant, minimally invasive surfactant therapy (MIST), respiratory distress syndrome, prematurity

## Abstract

Less-invasive surfactant administration (LISA), a newer technique of delivering surfactant via a thin catheter, avoids mechanical ventilation. LISA has been widely adopted in Europe but less so in the US. Our goal was to increase the percentage of surfactant delivered via LISA from 0% to 51% by 12/2020. Project planning and literature review started 12/2019, and included a standardized equipment kit and simulation training sessions. We began Plan–Do–Study–Act (PDSA) cycles in 6/2020. Initial exclusions for LISA were gestational age (GA) <28 weeks (w) or ≥36 w, intubation in the delivery room, or PCO2 >70 if known; GA exclusion is now <25 w. From 6 to 12/2020, 97 patients received surfactant, 35 (36%) via LISA. When non-LISA-eligible patients were excluded, 35/42 (83%) received LISA successfully. There were only 2/37 patients for whom LISA was not able to be performed. Three LISA infants required mechanical ventilation in the first week of life. Sedation remained an initial challenge but improved when sucrose was used routinely. LISA was safely and successfully introduced in our NICU.

## 1. Introduction

Less-invasive surfactant administration (LISA) is a technique to administer surfactant via a thin catheter. Using a thin catheter avoids positive pressure ventilation and allows the infant to remain on non-invasive respiratory support. Surfactant therapy has fundamentally altered the field of neonatology, increasing survival and decreasing viable gestational ages (GA) [1]. Early surfactant administration is associated with lower risk of air leak syndromes, reduced need for patent ductus arteriosus (PDA) treatment and bronchopulmonary dysplasia [1]. However, surfactant is not without risks. Most surfactant administration techniques require mechanical ventilation or the use of positive pressure ventilation (PPV). Even brief PPV, such as the intubation–surfactant–extubation (InSurE) technique, can lead to an inflammatory cascade that is associated with bronchopulmonary dysplasia (BPD) [2,3]. Aggressive weaning of ventilator settings and trials of continuous positive airway pressure (CPAP) prior to intubation have been used to avoid the drawbacks of mechanical ventilation [4]. This leads to a binary decision tree in which an infant is either exposed to early surfactant with barotrauma or surfactant is delayed following a trial of CPAP [5].

LISA techniques remove the binary decision by combining early surfactant with avoidance of PPV. Both CPAP and mechanical ventilation with surfactant were shown to have increased mortality and BPD rates compared to LISA, suggesting that LISA is useful as a lung-protective strategy [6,7,8,9]. InSurE uses a standard endotracheal tube (ETT) for surfactant administration which is then removed. Due to the ETT obstructing the glottis, the infant is unable to breathe physiologically during InSurE and requires PPV [10]. LISA uses an extremely thin catheter for surfactant administration, which allows the infant to breath physiologically [10]. This seemingly minor difference between InSurE and LISA may have significant consequences; compared to InSurE, LISA has improved mortality for all gestational ages [5,7]. Although most infants born at 22–24 weeks GA still require intubation, those who received LISA first had reduced rates of BPD and lower mortality compared to those who were mechanically ventilated. In addition to the proposed pulmonary benefits, there is emerging evidence of reduced rates and severity of intraventricular hemorrhage and a reduction in retinopathy of prematurity (ROP) compared to intubation and mechanical ventilation [11,12]. Due to the potential benefits and minimal risks, the European consensus on respiratory distress syndrome states “LISA is the preferred mode of surfactant administration for spontaneously breathing babies on CPAP, provided clinicians are experienced with this technique” [13].

LISA was first introduced in 1980 in the setting of low-resource areas aiming to reduce CPAP failure. This practice is widespread in Europe; as of 2017, 52% of European neonatologists were using LISA and 41% regarded it as the standard procedure for surfactant administration [14]. Since 2015, more surfactant is given with the LISA method than through ETT in hospitals of the German Neonatal Network [10]. Neonatologists within the United States have been slower to adopt LISA, with only 15% of institutions currently using LISA in any manner [15]. This may be due to lack of familiarity with the techniques for LISA. Equipment used for LISA includes placement of a feeding tube using Magill forceps, similar to the technique for nasotracheal intubation, use of a rigid angiocatheter placed directly in the trachea, and LISA-specific catheters. LISA-specific catheters include LISAcath (Chiesi Farmaceutici S.p.A., Parma, Italy) and Surfcath (Vygon, Ecouen, France); neither of which are available in the United States at the time of this publication. Placement of a rigid catheter for LISA was rated equivalent in difficulty to standard intubation techniques [16].

With the introduction of any new technique, safety is a concern. Complications from LISA are rare and include desaturation, surfactant reflux, bradycardia and apnea [17]. Most complications from LISA are minor, with desaturation requiring temporary increases in FiO_2_ or stimulation to recover. Complications remain rare even when controlling for GA [18]. One randomized controlled trial showed no difference in neurodevelopmental outcome (Bayley II scores) between LISA and standard intubation techniques [18].

Although technique and safety are well established, optimal sedation and analgesia for the procedure remain controversial aspects of LISA. The ideal regimen for any airway manipulation would be quick onset, short duration, and without respiratory depression. There is no medication that has these ideal properties; thus, a variety of strategies have been employed. Some centers use no sedation, citing the need for physiologic breathing for LISA to effectively distribute surfactant and the respiratory depression associated with sedation, other centers have used a variety of medication regimens from opioids to ketamine [15,19]. There is no clear consensus regarding appropriate sedation and analgesia for this procedure.

The potential for improved outcomes with minimal risk using LISA made the adoption of this therapy ideal to address our Neonatal Intensive Care Unit (NICU) patient population with respiratory distress syndrome requiring surfactant therapy without the need for mechanical ventilation. Due to the evidence for LISA and the barriers identified as unfamiliarity with the technique and lack of process related to administration, the authors decided that quality improvement methodology was the best method to implement LISA at our institution.

## 2. Materials and Methods

This project was reviewed by the University Hospitals Institutional Review Board and was determined to be non-human subjects research. Rainbow Babies and Children’s hospital NICU is an 82-bed level IV NICU divided between intensive care and transitional care units. The NICU receives 1500 admissions annually, of which 150 are very low birth weight (VLBW) infants. A variety of staff participate in the care of our infants including attending neonatologists, neonatal-perinatal medicine fellows, pediatric residents and neonatal nurse practitioners. Attending neonatologists and fellows are highly skilled in intubation techniques but comfort and skill level with intubation can vary among residents and neonatal nurse practitioners.

Based on the literature supporting LISA, we aimed to increase the use of LISA at Rainbow Babies and Children’s Hospital using the Model for Improvement. A multidisciplinary team including attending neonatologists, neonatal-perinatal medicine fellows, neonatal pharmacists, respiratory therapists, neonatal nurse practitioners and bedside nurses was formed. The four key drivers identified were patient selection, equipment selection, provider competence and buy in, and medication/sedation (Figure 1).

The SMART aim (which should be Specific, Measurable, Achievable, Relevant, and Timely) of our project was to increase the number of doses of surfactant administered by LISA in patients admitted to the NICU requiring surfactant replacement therapy from 0% to 51% by December 2020. The goal of 51% of surfactant administration was chosen based on pre-implementation data: 51% of infants who received surfactant via endotracheal tube prior to implementation of LISA were extubated within 48 h of surfactant administration. We believed that this population represented infants who required surfactant therapy without the need for additional ventilation support, and would therefore be ideal candidates for LISA.

Initial patient selection included infants born at 28 to 34 weeks of GA with a need for surfactant replacement therapy as evidenced by an FiO_2_ of 30% or greater (Figure 2). This selection of patients was chosen due to the perception that these patients would be at lowest risk of complications or acute respiratory failure. This allowed clinicians to gain experience and confidence in the technique and its therapeutic benefits prior to progressing to patients born at lower GA. We also initially excluded patients of higher gestational age due to the broad differential diagnosis for respiratory distress in these patients as well as the low risk of BPD. Additional criteria included spontaneously ventilating infants without significant apnea, pCO2 less than 70 on blood gas if obtained, and chest X-ray consistent with the primary diagnosis of respiratory distress syndrome (RDS), if obtained. We excluded patients who were intubated in the delivery room or prior to admission.

Due to the similarity and overlap between traditional intubation techniques and the use of a rigid catheter for LISA, our team selected a 16-gauge 5.5 inch angiocatheter (BD angiocath, Sandy, Utah) based on ease of use, cost and availability. We created a kit containing all necessary equipment to be stored in two areas of our NICU for easy access. The kit included sterile catheters, sterile markers, sterile measuring tape and sterile gloves.

We started with the population of caregivers most experienced in endotracheal intubation, including our neonatal-perinatal medicine fellows and attending neonatologists. Individual training sessions were completed with each provider on a training mannequin with a member of the LISA team.

For intubation at our institution, we use pre-medication (atropine, fentanyl and rocuronium) for all non-emergent intubations outside of the delivery room, due to improved patient comfort and a decrease in adverse events with premedication [20]. Due to controversy within the group of attending neonatologists, the plan for pharmacologic analgesia or sedation for LISA allowed for provider discretion. After several PDSA cycles, all patients were given comfort measures including swaddling and sucrose solutions. Choice of pharmacologic comfort measures, if desired, was left to the discretion of the provider preforming the procedure.

Data were obtained via the electronic medical record (EMR) as well as by data recorded by the provider team filled out at the time of procedure. Our primary process measure was administration of surfactant via LISA. Additional outcome measures included number of ventilator days, mortality, and BPD. Balancing measures included LISA attempted but failed, subsequent intubation within 24 h, and complications during LISA.

### Statistical Analysis

The primary outcome measure was percentage of surfactant doses given via LISA, grouped by month and plotted on a run chart. Wilcoxon test and Fisher’s exact test were used for statistical analysis between those who received surfactant by endotracheal tube and those who received surfactant by LISA.

## 3. Results

### 3.1. Interventions and PDSA Cycles

We performed an initial literature review and developed a guideline for the introduction of LISA. We obtained feedback from the division of neonatology and chose to limit initial GA at birth to 28 to 34 6/7 weeks. Over time, our initial population was expanded via PDSA cycles. First, we expanded the FiO2 requirement to any amount >21% with clinical symptoms of RDS and need for surfactant replacement therapy. This change was based on several infants requiring intubation after waiting for a period of time for them to reach an FiO_2_ of 30% prior to performing LISA. This changed allowed the therapy to be administered earlier, thus avoiding clinical decompensation.

We employed many rapid-cycle tests of change to determine the ideal regimen for analgesia or sedation prior to LISA. We found that infants greater than 24 h of age and with gestational age greater than 32 weeks were more likely to receive sedation or require additional LISA attempts. Therefore, these patients were identified as higher risk to inform the providers of potential sedation needs in this population. We found that familiarity with the technique improved over time and many infants did not require sedation for successful LISA procedure.

As planned, after successfully performing LISA in infants born at GA of 28 to 35 weeks in 10 patients, we lowered the GA to 25 weeks or greater. After these ten patients, we felt practitioners had the adequate skill level and there were minimal complications during the LISA procedure, and therefore we felt confident decreasing the GA eligibility.

### 3.2. LISA Success

From June to December 2020, 97 patients received surfactant and 35 (35%) of those patients received surfactant via the LISA technique (Figure 3). The median LISA usage increased from 0% at baseline (prior to the introduction of LISA) to 45% by December 2020, with six points above the median line noting a signal (Figure 4). The average gestational age of any infant receiving surfactant was 31.7 weeks (Table 1) and there was no difference between the gestational age or weights of infants who received surfactant by LISA vs. ETT (Table 1). Of the 37 infants who had LISA attempted, 8 had a birth weight less than 1500 g.

### 3.3. LISA Complications

There were no deaths among patients who received surfactant via LISA. Fifty-four percent experienced a complication during the procedure (Table 2). The most common complication was surfactant reflux (23%), or desaturation (14%). Most complications were short lived and required brief, temporary increases in oxygen during the procedure and rarely PPV. One infant was intubated during the procedure due to apnea and bradycardia. There was a single instance of failed LISA that necessitated intubation within 24 h for surfactant treatment. The average time on mechanical ventilation for infants who received LISA or in which LISA was attempted was 6 h vs. 81 h in infants who received surfactant via ET tube (*p* < 0.001, by Wilcoxon rank sum test). There was a significant drop in oxygen requirement following LISA from 39% to 26% (*p* < 0.0001, by Student’s *t*-test).

### 3.4. Bronchopulmonary Dysplasia (BPD) Outcomes

This QI initiative is part of an ongoing unit-wide goal of reducing BPD [9]. Using the NIH 2001 definition of BPD, gestational age of less than 32 weeks at birth and oxygen support at 36 weeks corrected gestational age or discharge [21], there were 50 very low birth weight (VLBW) infants who meet these criteria for BPD (Table 3). Only 4/20 patients (20%) who received LISA met criteria for mild BPD and no LISA candidates met criteria for moderate or severe BPD. This is in contrast to those who received ETT surfactant which had 17/24 (71%) meet criteria for BPD. However, our protocol was biased to provide LISA for babies of higher GA and reduced severity of illness. The infants who met criteria for BPD and received surfactant via ETT surfactant had a significantly lower gestational ages (26.1 weeks (ETT group) vs. 29.8 weeks (LISA group), *p* < 0.004 by Wilcoxon rank sum test) and birth weights (800 g vs. 1355 g, *p* < 0.0003 by Wilcoxon rank sum test) when compared with LISA candidates. In univariate analysis (Table 3) of VLBW infants, there were important differences between the two groups (Table 3). This study is not powered to determine BPD outcomes and data are included as BPD is a measure for our quality improvement initiative (Figure 5); however we recognize the LISA and non-LISA babies are quite different in their risk for BPD.

## 4. Discussion

### 4.1. LISA Usage

Using QI methodology, we were able to successfully and safely implement LISA at our level IV NICU. The median use of LISA increased from 0% of all first doses of surfactant given prior to introduction to a median of 45% by December 2020. This represents a substantial increase in the number of infants receiving surfactant via the LISA technique but falls short of our goal of 51%. This goal of 51% of surfactant given via LISA is supported both by our internal data and centers from the German neonatal network, where 50–55% of surfactant is given via the LISA technique [7]. There are a multitude of areas in which there are opportunities to increase the availability of LISA to meet our goal.

An area for improvement includes reduction in delivery room intubations as intubation prior to arriving in the unit was responsible for 60% of the infants who were not eligible for LISA. LISA has been used as an adjunct to neonatal resuscitation in spontaneously breathing infants with adequate heart rate and FiO_2_ requirements. This would have the additional benefit of early surfactant administration without delays due to transfer to the NICU. Of the centers practicing LISA 56% use LISA in some form in the delivery room. There is evidence that early LISA could prevent worsening respiratory status and subsequent intubation [22]. An uncontrolled environment is the cited as the reason other centers do not use LISA in the delivery room and fit with our justification to start our LISA experience within the controlled environment of our NICU. Future directions include administration of LISA as an adjunct to delivery room resuscitation while adhering to the neonatal resuscitation program (NRP) guidelines.

Following intubation in the delivery room, gestational age below our protocol lower limit was the second most common reason to not receive LISA. Based on our growing experience with LISA, we were able to successfully expand our gestational age down to 25 weeks. This recent expansion will help contribute to our goal of reaching 51% of surfactant given by LISA and will also be more effective in decreasing BPD. LISA has been used as low as 22 weeks, and most infants who receive LISA between 22 and 24 weeks will require intubation at some point. Although expansion below 25 weeks may not improve rates of any mechanical ventilation exposure, there still may be a BPD reduction benefit in this population. We will have to continue to monitor our complications and outcome measures as we approach these lower gestational ages.

### 4.2. Performance

We were able to successfully train all of our fellows on LISA, including first year fellows who had varying levels of experience with intubation. None of our providers had prior experience or training with this technique. Training on a mannequin was essential to our successful initiation as it allowed practitioners to understand the similarities and differences between intubation and LISA and anticipate the experience during a real procedure, including marking the catheter with a black marker. Based on our high success rate of LISA procedures with fellows, we are expanding LISA training to other airway providers, starting with neonatal nurse practitioners.

When appropriately applied 82% of infants that met criteria for LISA underwent the procedure, with only 5 infants being excluded for provider discretion. As attending neonatologists could elect to use LISA or not without justification, this represents high provider buy-in and confidence in the procedure. Our experience has shown that centers without LISA experience can institute LISA successfully.

### 4.3. Clinical Outcomes

There was a significant decrease in the use of mechanical ventilation since the implementation of LISA. This reduction in mechanical ventilation suggests appropriate patient selection based on their success. In our Vermont Oxford Network (VON) data obtained prior to LISA implementation, each quartile averaged between 10 and 15 infants who required intubation greater than 29 weeks. In our last quartile only 3 infants greater than 29 weeks required intubation, a significant drop in intubations for only surfactant administration.

This quality improvement (QI) initiative is part of an ongoing effort to reduce BPD, but this study is not powered for BPD outcomes at this time. Results are included as this outcome is part of our long-term LISA measures. The LISA and non-LISA surfactant groups are clearly different and with the large difference in median gestational age and birth weight. Most of the infants who received surfactant via ETT were intubated and given surfactant in the delivery room. This may be representative of a greater severity of illness in that population compared to the LISA population as the more clinically ill patients do not qualify for LISA. In this quality improvement study, we are not able to say that implementation of LISA was responsible for any changes BPD seen in our LISA population. Metanalyses and RCT with similar patient characteristics have shown a reduction in BPD [9]. We are continuing to work toward reduction in BPD using a multipronged unit-wide approach, of which LISA is only one component. Further data will be required to determine the effect, if any LISA has on BPD.

No infant who underwent LISA expired and no major safety events were associated with the use of LISA. Excluding failed LISA attempts, most of the complications seen were mild and self-limited. The most common complications of the procedure were surfactant reflux and desaturation. Even with the high percentage of surfactant reflux in the infants who received LISA, the patients still saw a significant decrease in oxygen use suggesting that surfactant reflux has minimal effect on the efficacy of LISA. Desaturations were mostly self-limited or responded to transient increases in oxygen during surfactant administration with subsequent ability to wean below the previous baseline of FiO_2_. Our data support an excellent safety record for LISA even in a center with minimal prior experience.

### 4.4. Sedation/Analgesia

Sedation or analgesia remains a controversial topic within the LISA literature. The dichotomy remains between ensuring patient comfort during the procedure without respiratory depression leading to failure of CPAP. We focused on analgesia with sucrose solution and non-pharmacologic management such as swaddling during the procedure, while allowing practitioners to make decisions based on their clinical judgement and the individual patient condition. The vast majority of our LISA procedures were performed with only sucrose solution and other non-pharmacologic measures. A survey of our staff found that the majority want to attempt LISA without additional pharmacologic sedation while having a plan for sedation in the case of additional attempts. We are continuing to monitor infants’ responses to LISA to develop further guidance in regard to pharmacologic comfort measures.

## 5. Conclusions

LISA can be successfully and safely implemented at a center without prior experience in the technique. There was a significant decrease in mechanical ventilation hours compared to those who did not receive LISA. Complications were mostly mild and required only minimal intervention and there were no major safety events surrounding the LISA procedure. Future directions include continuing to test and implement strategies to reach our goal of 51% of surfactant being administered via the LISA technique. Next areas of focus include reducing delivery room intubations, introducing LISA to the delivery room as an adjunct in resuscitation, and expanding this valuable therapy to all gestational ages.

## Figures and Tables

**Figure 1 children-08-00580-f001:**
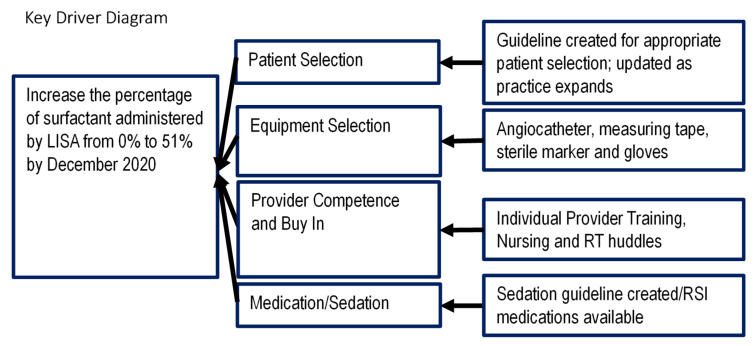
Key drivers for implementation of LISA to our unit. RT, respiratory therapist; LISA, less invasive surfactant administration.

**Figure 2 children-08-00580-f002:**
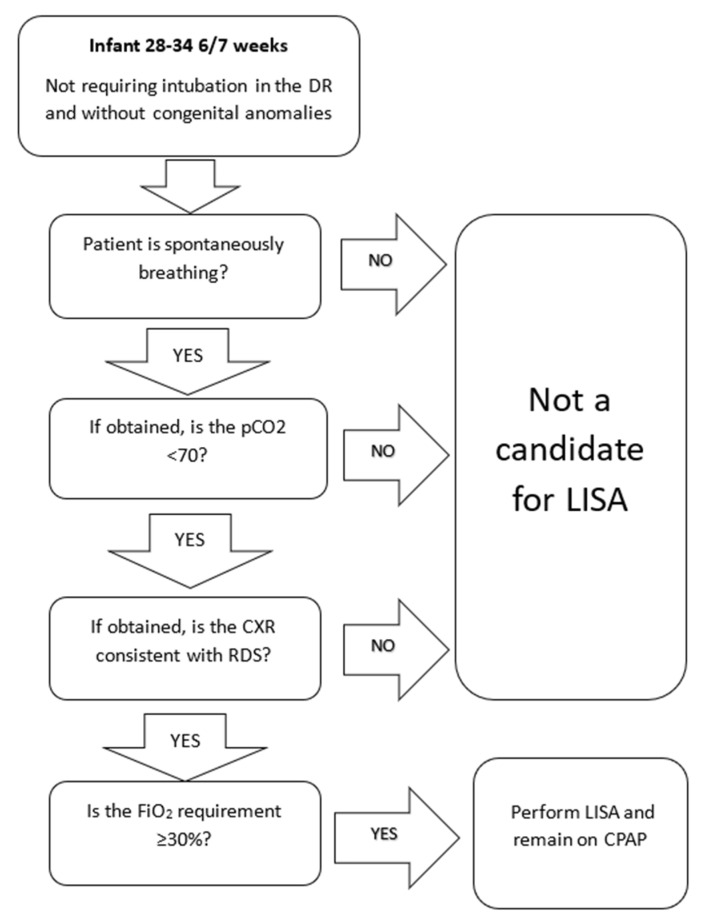
Flow chart demonstrating initial selection of patients for LISA. This charted aided in selecting infants who were spontaneously ventilating, required surfactant administration but did not require mechanical ventilation. FiO_2_ requirement was later reduced to ≥22% during a PDSA cycle and we are now including babies 25–35 6/7 weeks. LISA, less-invasive surfactant administration, PDSA, plan-do-study-act, DR, delivery room, CXR, chest X-ray, CPAP, continuous positive airway pressure, RDS, respiratory distress syndrome.

**Figure 3 children-08-00580-f003:**
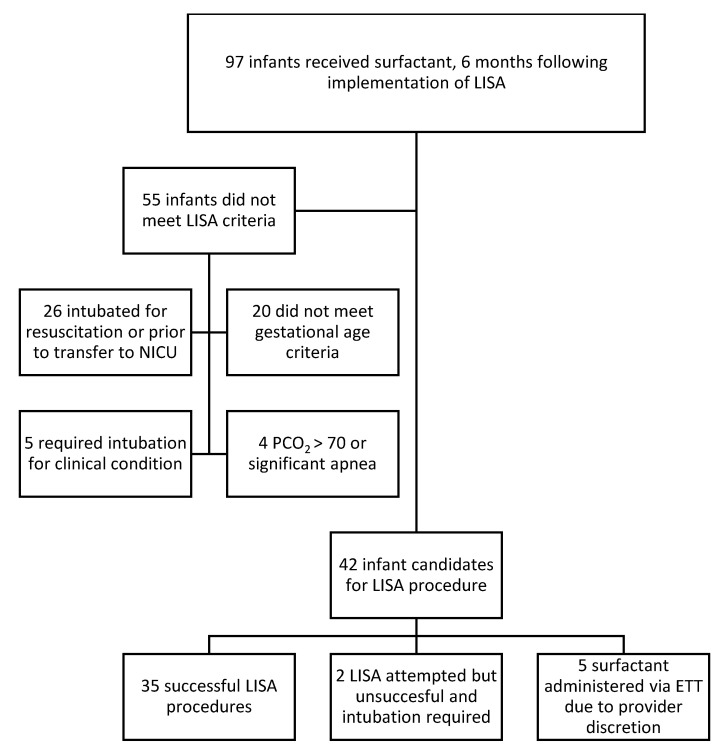
Surfactant administration. The most common reason for not meeting LISA eligibility was intubation in the delivery room or prior to transfer/admission. Five had a clinical condition such as congenital diaphragmatic hernia or pneumothorax that necessitated intubation. Providers had discretion to not perform LISA even if the patient otherwise met criteria and this occurred with five infants. ETT, endotracheal tube.

**Figure 4 children-08-00580-f004:**
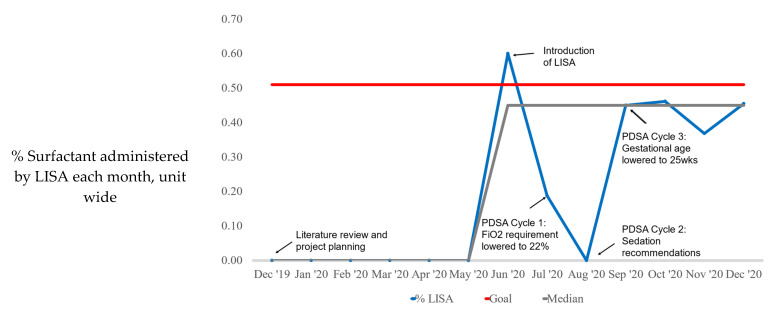
Annotated run chart: percentage of infants who received surfactant via LISA with associated PDSA cycles. There were no infants eligible in August due to intubation prior to transfer or gestational age below our protocol criteria.

**Figure 5 children-08-00580-f005:**
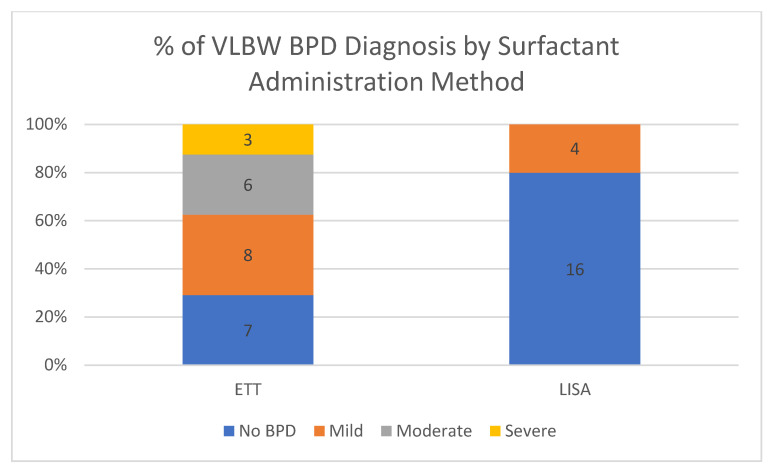
BPD severity by grade based on ETT or LISA for surfactant administration. BPD, bronchopulmonary dysplasia.

**Table 1 children-08-00580-t001:** Characteristics of Infants Who Received Surfactant via ETT vs. Attempted LISA.

	ETT Surfactant	LISA *	*p*-Value
Number of Patients	60	37	
GA (weeks)	32.2	31.0	0.4298
Weight (grams)	1770	1765	0.92
% Male	65%	57%	0.51
Mechanical Ventilation (1st week of life)	100%	8%	0.0001
Duration of Mechanical Ventilation (hours)	52.5	0	0.0001
Mortality	10%	0%	0.079

Data are presented as the median (tested using Wilcoxon rank-sum) or by percent (tested using Fisher’s exact test). There was significantly less mechanical ventilation in the 1st week of life and longer duration of mechanical ventilation with LISA. * includes all babies who had LISA attempted (2/37 were unsuccessful).

**Table 2 children-08-00580-t002:** LISA Complications.

Complications during LISA Procedure	
All babies who received LISA	35 (100%)
Total with a complication	19 (54%)
Surfactant Reflux	9 (25%)
Desaturation, saturation less than 60% during procedure	6 (17%)
Apnea	5 (14%)
Intubation within 24 h	2 (5%)
Cough	1 (3%)

Complications associated with LISA procedure. One infant required immediate intubation for apnea and bradycardia. Most complications required only supportive care or a transient increase in FiO_2_ with subsequent decrease within 1 h. Some babies are represented in more than one category.

**Table 3 children-08-00580-t003:** Characteristics of Very Low Birth Weight Infants <32 Weeks.

VLBW Infants <32 Weeks	ETT	LISA	*p* Value
Number	30	20	N/A
Weight (g)	800	1355	0.0003
Gestational Age (weeks)	26.1	29.8	0.004
Mortality	20%	0%	0.069
Sex (% male)	57%	60%	1
Any Mechanical Ventilation	100%	0%	0.0001
Duration of Mechanical Ventilation (h)	136.5	0	0.0001

Data are presented as the median (tested using Wilcoxon rank sum) for variables not normally distributed or by percent (tested using Fisher’s exact test). This table includes infants whose gestational age at birth was less than 32 weeks and corrected gestational age was 36 weeks by December 2020 (so that BPD outcome was available). Infants who received surfactant by ETT had significantly lower weight and gestational age.

## Data Availability

The data presented in this study are available on request from the corresponding author.

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
