# Peer review of "Introducing Less-Invasive Surfactant Administration into a Level IV NICU: A Quality Improvement Initiative"

_children, 2021, doi:10.3390/children8070580_

Round 1
Reviewer 1 Report
This is a manuscript about the implementation of LISA in a Level 4 NICU in the USA.
As the uptake of LISA is not yet established in the USA, this paper is of interest.
By the team well-thought en careful steps were taken in the implementation, which improves the chances of succesful implementation.
However i have some concerns..
1. Most importantly, this study is not designed for impact on the major outcomes (BPD etc) and certainly not for comparison with the intubated group as this involves bias. Although this was discussed, it is not sound to present these results, due to power and bias concerns. Conclusion can not be drawn from this and must be left out. I would advise the authors just to focus on succesful implementation and only mention short term outcome measures such as need for ventilation.
2. Figure 4 is not clear. What is the unit of %, per month/per week? Why is there a large decrease in August?
3. I agree with the restriction on GA at start of LISA. However i notice only 13 were excluded because of GA (below 28 weeks and later 25? That number seems very small to me in a level IV NICU. Can the authors be more precise in the flow chart. And show all infants who received surfactant and than subdivide which infants got surfactant after LISA, which were intubated for surfactant and elaborate on the infants (n=33) who already received surafctant (seems rather high).
4. Sedation for LISA is an important topic. I would like to compliment the team for their focus on non-pharmalogical pain reduction/comfort. However I have concerns whit the use of midazolam, which suppresses the respiratory drive, maybe even more than fentanyl. I would advise the authors to restrict the use of sedatives to the ones that have been investigated, or at least comment on these studies (for example propofol).
Author Response
Reviewer 1
This is a manuscript about the implementation of LISA in a Level 4 NICU in the USA.
As the uptake of LISA is not yet established in the USA, this paper is of interest.
By the team well-thought en careful steps were taken in the implementation, which improves the chances of succesful implementation.
We thank the reviewer for his/her positive comments and for the opportunity to address concerns. Yes we have found that although several other NICU groups here in the US are talking about using LISA, most of them have not done so at this time.
However i have some concerns..
- Most importantly, this study is not designed for impact on the major outcomes (BPD etc) and certainly not for comparison with the intubated group as this involves bias. Although this was discussed, it is not sound to present these results, due to power and bias concerns. Conclusion can not be drawn from this and must be left out. I would advise the authors just to focus on successful implementation and only mention short term outcome measures such as need for ventilation.
We agree there is tremendous selection bias in the LISA vs. non-LISA group. However, we included some other outcomes because we were often asked for this data in other forums, and since that is actually our long term goal to impact BPD. We have reworded these findings in text/paragraph only with no p values, simply stating the outcome and with the caveats already previously stated. If the reviewer would like us to remove completely the BPD outcome we have no objection to do so. We have made the following alterations to remove direct conclusions about our use of LISA and BPD.
We removed from the abstract line 21-22 “Our global aim is to continue to work toward reducing BPD for the highest risk lower GA patients”
Line 43-45 the line was changed to less definitive language “suggesting that LISA is useful as a lung protective strategy”
Line 49: “This seemingly minor difference between InSurE and LISA has significant consequences; compared to InSurE, LISA has improved mortality for all gestational ages [5, 7]” was changed to “This seemingly minor difference between InSurE and LISA may significant consequences; compared to InSurE, LISA has improved mortality for all gestational ages [5, 7]”
Line 51: “In infants greater than 28 weeks GA at birth, the use of LISA reduced rates of intubation and oxygen use. In infants born at 22-27 weeks GA at birth there is a reduction in BPD and mechanical ventilation with LISA [7].” Was removed
Line 93: was changed from “LISA’s proven superiority to other surfactant administration techniques, record of safety, and similarity to intubation techniques” was changed to “LISA’s potential for improved outcomes with minimal risk made the adoption of this therapy ideal to address our Neonatal Intensive Care Unit (NICU) patient population with respiratory distress syndrome requiring surfactant therapy without the need for mechanical ventilation
Line 100: “Due to the strong evidence for LISA and the barriers; identified as unfamiliarity with the technique and lack of process related to administration, the authors decided that quality improvement methodology was the best method to implement LISA at our institution., thus working toward our global aim of reducing BPD for premature infants.” The word strong was omitted as well as the goal of reducing BPD thus changed to “Due to the evidence for LISA and the barriers; identified as unfamiliarity with the technique and lack of process related to administration, the authors decided that quality improvement methodology was the best method to implement LISA at our institution.”
Line 325 “ Only 4 VLBW infants who received LISA were diagnosed with BPD based on the defini-tion of supplemental oxygen use at 36 weeks postmenstrual age (PMA); this rate was significantly lower compared with ETT surfactant in the VLBW population. “was removed and replaced with “This QI initiative is part of an ongoing unit wide effort to reduce BPD via QI, but this study is not powered for BPD outcomes at this time. Results are included as BPD out-comes is part of our long-term LISA measures.”
Line 331 was removed “, it was not surprising that multiple regression analysis showed that LISA was not independently predictive of a reduction in BPD.”
Line 337 was changed from “In this quality improvement study, we are not able to say that implementation of LISA was responsible for the lower rates BPD seen in our LISA population. but we have a small sample size less than 1500 grams. To “In this quality improvement study, we are not able to say that implementation of LISA was responsible for any changes BPD seen in our LISA population”
Line 341 added further data will be required to determine the effect, if any LISA, has on BPD.
Line 375 removed, along with our global aim to reduce bronchopulmonary dysplasia in our unit
- Figure 4 is not clear. What is the unit of %, per month/per week? Why is there a large decrease in August?
Figure 4 is a run chart depicting the % of surfactant give via LISA per month. The title was changed to “% Surfactant Administered by LISA each month, Unit Wide” The decrease in August is due to no eligible infants in August, due to gestational age criteria or due to intubation in the delivery room. The following was added to the caption for figure 4 “There were no infants eligible in August due to intubation during resuscitation or gestational age below our protocol criteria.”
- I agree with the restriction on GA at start of LISA. However i notice only 13 were excluded because of GA (below 28 weeks and later 25? That number seems very small to me in a level IV NICU. Can the authors be more precise in the flow chart. And show all infants who received surfactant and than subdivide which infants got surfactant after LISA, which were intubated for surfactant and elaborate on the infants (n=33) who already received surafctant (seems rather high).
Of the 33 who received surfactant, 7 of those had a gestational age lower than our protocol and were moved to the gestational age category. Thus 26 were intubated per NRP guidelines in the delivery room, while 20 would not have met criteria due to their gestational age. There is another QI project ongoing at our hospital to reduce our rates of delivery room intubations. We hope in the future that our LISA project can be adjunct to our NRP resuscitative efforts.
- Sedation for LISA is an important topic. I would like to compliment the team for their focus on non-pharmalogical pain reduction/comfort. However I have concerns whit the use of midazolam, which suppresses the respiratory drive, maybe even more than fentanyl. I would advise the authors to restrict the use of sedatives to the ones that have been investigated, or at least comment on these studies (for example propofol).
We agree this is an important issue and we do not have the perfect solution. After the first ten babies, we have not had any further use /need for sedation. This may be everyone being more comfortable with the procedure and not using our standard triad of pre-intubation medications. We are not advocating midazolam. We removed the line 193 “Midazolam and fentanyl were chosen as our recommended medications based on their short half-life, favorable side effect profile and our institutional experience with these medications.”
Additionally, we reworded line 162 to “Choice of pharmacologic comfort measures, if desired, were left to the discretion of the provider preforming the procedure.” To reflect the varying preferences for sedation in our unit and the lack of definitive evidence for any sedation techniques.
Reviewer 2 Report
For my point of view this is a useful manuscript for scholars as it describes a safe way to implement LISA in a NICU which does not use it.
My objections arise for the way that it is written and especially for their results referred to BPD. In the manuscript in general they described an absolute correlation of LISA with BPD.
In a very recent paper (2021) in Acta Paediatrica by Mehler K et al conclusions are drawn “LISA is safe and may be superior”. In the same paper babies <27 w GA did not present significant difference in the overall rate of BPD (p=0.112) and the difference found (p=0.010) in babies 25/26 w GA was also difference in GA (p=0.010) and BW (p=0.006) between the groups.
In JAMA 2016, Isiyama et al concluded about the same issue “these findings were limited by the overall low quality of evidence and lack of robustness in higher quality trials”.
It is a mistake to refer in the abstract “our global aim is to continue to work toward reducing BPD”, describing LISA application and comparing in the manuscript very preterm neonates with BW 800g and 1355g.
Authors also stated in introduction about the issue “there is no clear consensus regarding appropriate sedation and analgesia for this procedure”. It is obvious that not everything is yet clear.
So, I believe for all the above reasons, authors have to remove the last sentence from the Abstract and to rewrite parts of the Introduction and the Discussion, considering and using one less causal language.
The self-evident truth that LISA applied with knowledge and precision causes minimal inconvenience (handling) compares with ETT in the very tiny babies, it is reason enough to promote its wider application.
Author Response
For my point of view this is a useful manuscript for scholars as it describes a safe way to implement LISA in a NICU which does not use it.
We thank the reviewer for this positive comment. Yes we have found that although several other NICU groups here in the US are talking about using LISA, most of them have not done so at this time.
My objections arise for the way that it is written and especially for their results referred to BPD. In the manuscript in general they described an absolute correlation of LISA with BPD.
We agree there is tremendous selection bias in the LISA vs. non-LISA group. However, we included some other BPD outcomes because we were often asked for this data in other forums, and since that is actually our long term goal (to impact BPD). We have reworded these findings in text/paragraph only with no p values, simply stating the outcome and with the caveats already previously stated. If the reviewer would like us to remove completely the BPD outcome we have no objection to do so.
In a very recent paper (2021) in Acta Paediatrica by Mehler K et al conclusions are drawn “LISA is safe and may be superior”. In the same paper babies <27 w GA did not present significant difference in the overall rate of BPD (p=0.112) and the difference found (p=0.010) in babies 25/26 w GA was also difference in GA (p=0.010) and BW (p=0.006) between the groups.
In JAMA 2016, Isiyama et al concluded about the same issue “these findings were limited by the overall low quality of evidence and lack of robustness in higher quality trials”.
While we are aware of the limitations Dr. Isayama stated in his meta-analysis, there are several other meta-analyeses [Rigo 2017, Aldana-Aguirre 2017, ] that also conclude that LISA appears superior for several outcomes, including BPD. Despite his caveat, his paper is quite compelling to us, and likely compelling to the editors at JAMA. Even so we have made the following alterations to remove any absolute correlation of BPD and LISA.
We removed from the abstract line 21-22 “Our global aim is to continue to work toward reducing BPD for the highest risk lower GA patients”
Line 43-45 the line was changed to less definitive language “suggesting that LISA is useful as a lung protective strategy”
Line 49: “This seemingly minor difference between InSurE and LISA has significant consequences; compared to InSurE, LISA has improved mortality for all gestational ages [5, 7]” was changed to “This seemingly minor difference between InSurE and LISA may significant consequences; compared to InSurE, LISA has improved mortality for all gestational ages [5, 7]”
Line 51: “In infants greater than 28 weeks GA at birth, the use of LISA reduced rates of intubation and oxygen use. In infants born at 22-27 weeks GA at birth there is a reduction in BPD and mechanical ventilation with LISA [7].” Was removed
Line 93: was changed from “LISA’s proven superiority to other surfactant administration techniques, record of safety, and similarity to intubation techniques” was changed to “LISA’s potential for improved outcomes with minimal risk made the adoption of this therapy ideal to address our Neonatal Intensive Care Unit (NICU) patient population with respiratory distress syndrome requiring surfactant therapy without the need for mechanical ventilation
Line 100: “Due to the strong evidence for LISA and the barriers; identified as unfamiliarity with the technique and lack of process related to administration, the authors decided that quality improvement methodology was the best method to implement LISA at our institution., thus working toward our global aim of reducing BPD for premature infants.” The word strong was omitted as well as the goal of reducing BPD thus changed to “Due to the evidence for LISA and the barriers; identified as unfamiliarity with the technique and lack of process related to administration, the authors decided that quality improvement methodology was the best method to implement LISA at our institution.”
Line 325 “ Only 4 VLBW infants who received LISA were diagnosed with BPD based on the defini-tion of supplemental oxygen use at 36 weeks postmenstrual age (PMA); this rate was sig-nificantly lower compared with ETT surfactant in the VLBW population. “was removed and replaced with “This QI initiative is part of an ongoing unit wide effort to reduce BPD via QI, but this study is not powered for BPD outcomes at this time. Results are included as BPD out-comes is part of our long-term LISA measures.”
Line 331 was removed “, it was not surprising that multiple regression analysis showed that LISA was not independently predictive of a reduction in BPD.”
Line 337 was changed from “In this quality improvement study, we are not able to say that implementation of LISA was responsible for the lower rates BPD seen in our LISA population. but we have a small sample size less than 1500 grams. To “In this quality improvement study, we are not able to say that implementation of LISA was responsible for any changes BPD seen in our LISA population”
Line 341 added further data will be required to determine the effect, if any LISA, has on BPD.
Line 375 removed, along with our global aim to reduce bronchopulmonary dysplasia in our unit
It is a mistake to refer in the abstract “our global aim is to continue to work toward reducing BPD”, describing LISA application and comparing in the manuscript very preterm neonates with BW 800g and 1355g.
We eliminated this sentence.
Authors also stated in introduction about the issue “there is no clear consensus regarding appropriate sedation and analgesia for this procedure”. It is obvious that not everything is yet clear.
So, I believe for all the above reasons, authors have to remove the last sentence from the Abstract and to rewrite parts of the Introduction and the Discussion, considering and using one less causal language.
We do not understand your objection. We agree “not everything is yet clear” and that is why we said “there is no clear consensus” To remove any recommendations for specific medications. Additionally please see the above changes that were made to remove causal language.
The self-evident truth that LISA applied with knowledge and precision causes minimal inconvenience (handling) compares with ETT in the very tiny babies, it is reason enough to promote its wider application.
Thank you for this phrasing. We have tried to incorporate this “feeling” in our conclusion.
Round 2
Reviewer 2 Report
I have not any other suggestion for authors. I considered that this manuscript it will be useful for scholars and have to be published.
Author Response
Thank you for your comments and time to help improve this manuscript.